# Structural dynamics of single SARS-CoV-2 pseudoknot molecules reveal topologically distinct conformers

Krishna Neupane [1,3], Meng Zhao [1,3], Aaron Lyons[1], Sneha Munshi[1], Sandaru M. Ileperuma[1], Dustin B. Ritchie[1], Noel Q. Hoffer[1], Abhishek Narayan[1] & Michael T. Woodside [1,2 ✉]

The RNA pseudoknot that stimulates programmed ribosomal frameshifting in SARS-CoV-2 is a possible drug target. To understand how it responds to mechanical tension applied by ribosomes, thought to play a key role during frameshifting, we probe its structural dynamics using optical tweezers. We find that it forms multiple structures: two pseudoknotted conformers with different stability and barriers, and alternative stem-loop structures. The pseudoknotted conformers have distinct topologies, one threading the 5′ end through a 3-helix junction to create a knot-like fold, the other with unthreaded 5′ end, consistent with structures observed via cryo-EM and simulations. Refolding of the pseudoknotted conformers starts with stem 1, followed by stem 3 and lastly stem 2; $Mg^{2+}$ ions are not required, but increase pseudoknot mechanical rigidity and favor formation of the knot-like conformer. These results resolve the SARS-CoV-2 frameshift signal folding mechanism and highlight its conformational heterogeneity, with important implications for structure-based drug-discovery efforts.

[1] Department of Physics, University of Alberta, Edmonton, AB, Canada. [2] Li Ka Shing Institute of Virology, University of Alberta, Edmonton, AB, Canada. [3] These authors contributed equally: Krishna Neupane, Meng Zhao. ✉email: michael.woodside@ualberta.ca

like most coronaviruses, the Severe Acute Respiratory Syndrome coronavirus 2 (SARS-CoV-2) causing the COVID-19 pandemic makes use of −1 programmed ribosomal frameshifting (−1 PRF) to express proteins that are essential for viral replication[1]. In −1 PRF, a shift in the reading frame of the ribosome at a specific location in the RNA message is stimulated by a structure in the mRNA located 5–7 nt downstream of the slippery sequence where the reading frameshift occurs, thereby generating alternate gene products[2,3]. Previous work on viruses including HIV-1 and SARS-CoV showed that mutations modulating the level of −1 PRF can significantly attenuate viral propagation in cell culture[4–6]. As a result, the structures stimulating −1 PRF are potential targets for anti-viral drugs[7–9], motivating efforts to find ligands active against −1 PRF in SARS-CoV-2 that could be used to treat COVID-19[10–14].

The pseudoknot stimulating −1 PRF in SARS-CoV-2 has a three-stem architecture[1,10,15,16] (Fig. 1a) that is characteristic of coronaviruses, in contrast to the more common two-stem architecture of most viral frameshift-stimulatory pseudoknots[17]. Cryo-EM imaging[10,15] and computational modeling[18] both suggest that the SARS-CoV-2 pseudoknot can take on several different conformers (Fig. 1b, c). Some of these conformers involve knot-like fold topologies that have not previously been observed in frameshift-stimulatory pseudoknots, specifically conformers with the 5′ end threaded through the junction between the three helices to generate what we term a ring-knot[10,15,18]. Such a 5′-end threaded ring-knot fold has only previously been observed in viral exoribonuclease-resistant RNAs[19–21]. Intriguingly, the co-existence of multiple conformers in the SARS-CoV-2 pseudoknot is consistent with evidence from studies of various stimulatory structures, both pseudoknots and hairpins[22–25], as well as from studies of the effects of anti-frameshifting ligands[26], showing that the stimulation of −1 PRF is linked to conformational heterogeneity in the stimulatory structure. In particular, −1 PRF is linked to the conformational heterogeneity under tension[27] in the range of forces applied by the

ribosome during translation[28,29]. However, the dynamic ensemble of conformers populated by the SARS-CoV-2 pseudoknot has not yet been explored experimentally, and the folding mechanism of this pseudoknot—especially its unusual ring-knotted conformer—remains unknown.

Here we examine the conformational dynamics of the SARS-CoV-2 pseudoknot in the single-molecule regime. We study it under tension in optical tweezers in order to mimic the situation seen during −1 PRF, where the force applied by the ribosome is ramped up and down as the ribosome attempts to resolve the mRNA structure before shifting reading frame[30]. Such force spectroscopy measurements are also a powerful tool for characterizing folding mechanisms[31] and the energy landscapes that govern folding dynamics[32]. We find that the SARS-CoV-2 frameshift signal indeed forms at least two distinct pseudoknotted conformers, one involving threading of the 5′-end to form a ring-knot and the other without any threading. Stem 1 usually folds first, followed by stem 3 and lastly stem 2, but sometimes alternate stem-loops form that displace the pseudoknotted structures; $Mg^{2+}$ rigidifies the pseudoknot structures and favors the formation of the threaded conformer. The existence of multiple conformers of this frameshift signal has important implications for structure-based efforts to find small-molecule therapeutics targeting −1 PRF.

## Results

To probe the conformations formed by the SARS-CoV-2 pseudoknot, their folding pathways, and the dynamics under tension, we annealed a single RNA molecule containing the sequence of the pseudoknot flanked by handle regions to DNA handles that were attached to beads held in optical traps (Fig. 2a). We then moved the traps apart to ramp up the force and unfold the RNA, and brought them back together to ramp down the force and refold the RNA. Force-extension curves (FECs) measured during unfolding in near-physiological ionic conditions (130 mM $K^+$, 4 mM $Mg^{2+}$) showed one or more characteristic transitions in which the extension abruptly increased and force simultaneously decreased when part or all of the structure unfolded cooperatively (Fig. 2b). Unfolding events were observed over a range of forces from ~5 to 50 pN; similar transitions were seen in refolding FECs, but at forces below ~15 pN (Fig. 2c).

Examining the unfolding FECs in more detail, we found two qualitatively different behaviors distinguished by different length changes. Measuring the amount of RNA unfolded by fitting the FECs before and after the transition to worm-like chain (WLC) polymer elasticity models (Eq. (1), Methods) for the handles and unfolded RNA (Fig. 2b, dashed lines), we found that ~80% of FECs (Fig. 2b, black and magenta) showed a contour length change of $\Delta L_c = 35.6 \pm 0.4$ nm for complete unfolding (Supplementary Table 1). This result was consistent with the value expected for full unfolding of the pseudoknot, 34.7–36.5 nm, based on cryo-EM reconstructions[10,15] and computational modeling of likely structures[18]. Sometimes these curves contained an intermediate state, $I_1$ (Fig. 2b, magenta), which unfolded with a length corresponding to stem 1 ($\Delta L_c = 17.5 \pm 0.4$ nm), but most of the time they showed a single cooperative unfolding transition (Fig. 2b, black). The remaining ~20% of the FECs (Fig. 2b, green) unfolded with a smaller total length change of $\Delta L_c = 25 \pm 1$ nm, indicating an alternate structure that was incompletely folded (denoted as Alt), at forces of ~10–20 pN characteristic of hairpins[33] (Supplementary Fig. 1). Such alternative structures have been observed previously for many frameshift signals[22,34]. This second class of FECs also often contained at least one unfolding intermediate.

There is a characteristic shape expected for the distribution of unfolding forces, $p(F_u)$, for unfolding across a single barrier[35],

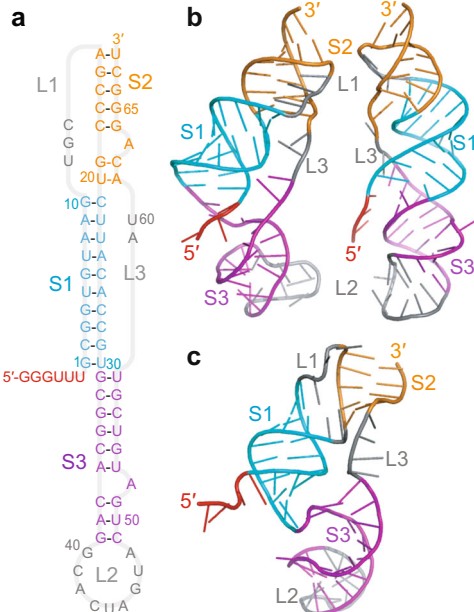

**Fig. 1 Three-stemmed architecture of SARS-CoV-2 frameshift-stimulatory pseudoknot. a** Secondary structure from Ref. [1], with stems and loops color-coded (S1: stem 1, cyan; S2: stem 2, orange; S3: stem 3, purple; L1/2/3: loop 1/2/3, gray). Spacer region linking to slippery sequence shown in red. **b** 3D structures of 5′-threaded (left) and -unthreaded (right) conformers from Ref. [18]. **c** Cryo-EM structure from Ref. [15] (PDB ID 7O7Z).

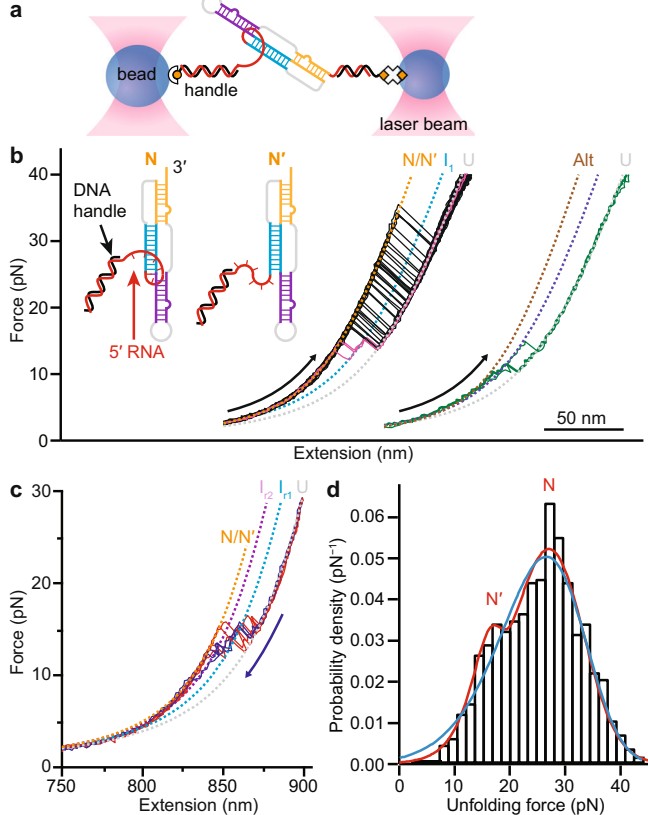

**Fig. 2 Force spectroscopy of SARS-CoV-2 pseudoknot. a** Schematic of force spectroscopy assay: a single RNA pseudoknot is tethered by duplex handles attached to beads held in optical traps. Pseudoknot secondary structure color-coded as in Fig. 1. **b** Unfolding force-extension curves (FECs) with 6-nt spacer between handle and pseudoknot (inset: schematic of conformers N and N′ showing handle location). Black: FECs showing full-length unfolding of pseudoknot without intermediates. Magenta: FECs unfolding pseudoknot via intermediate state $I_1$. Green: FECs showing unfolding of alternate stem-loop structures (labeled Alt). Dashed lines: WLC fits to each state (orange: N/N′, cyan: $I_1$, gray: U, brown: Alt; navy: intermediate for Alt). FECs offset for clarity. **c** FECs showing full-length refolding of pseudoknot, with intermediates corresponding to folding of stem 1 ($I_{r1}$, cyan dashed line) and stem 1 + stem 3 ($I_{r2}$, purple dashed line). **d** Unfolding force distribution for pseudoknot ($N = 745$ FECs from 10 molecules) shows two peaks, representing the distinct conformers N and N′. Blue: fit to single-population landscape model; red: fit to two-population landscape model.

hence $p(F_u)$ can reveal the presence of distinct initial states during the unfolding[36,37]. For the population of FECs with full-length unfolding, two peaks were seen in $p(F_u)$: a minor peak near 16 pN and a larger peak near 30 pN (Fig. 2d, black). The double peak indicates the presence of at least two distinct initial conformers, which despite sharing the same total length change nevertheless unfold over different barriers, leading to different shapes for their unfolding force distributions. Such behavior has been seen previously in ligand-bound riboswitches[38] and proteins[36]. By fitting $p(F_u)$ to a kinetic model for barrier crossing (Eq. 2, Methods), the shape of the energy barrier can be characterized through its height ($\Delta G^{\ddagger}$) and distance from the folded state ($\Delta x^{\ddagger}$), reporting on the nature of the interactions that hold the structure together[32]. We found that $p(F_u)$ fit better to the distribution expected for two initial states (Fig. 2d, red; fit parameters listed in Supplementary Table 2) than to the distribution expected for a single initial state (Fig. 2d, blue), as assessed by the Akaike

information criterion (AIC) (see Methods)[39]. The results for $\Delta G^{\ddagger}$ were similar within error for the two initial states, respectively, 31 ± 4 kJ/mol for the higher-force state (denoted N) and 33 ± 5 kJ/mol for the lower-force state (denoted N′), but $\Delta x^{\ddagger}$ was notably smaller for N: 0.7 ± 0.1 nm, compared to 2.1 ± 0.3 nm for N′, implying a more rigid structure for N. In both cases, however, the value for $\Delta x^{\ddagger}$ was consistent with the range characteristic of pseudoknots[22,40] and other structures containing tertiary contacts[41–43], but too short for structures consisting only of stem-loops[33]. The great majority of the unfolding FECs showing full-length $\Delta L_c$ started in state N (91 ± 2%), with only a small minority (9 ± 2%) starting in state N′.

Given that the SARS-CoV-2 pseudoknot is predicted to form different fold topologies, such as the 5′-threaded and -unthreaded conformers seen in simulations[18], such different conformers would be expected to give rise to sub-populations with different mechanical properties, because 5′-threaded folds are generally more mechanically resistant than unthreaded folds[21]. To test if the high-force population involved threading of the 5′ end, we explored if the proportions of the high-force and low-force populations could be modulated by changing the proximity of the duplex handle to the 5′ end of stem 1: steric hindrance from a bulky duplex that is too close to the stem 1/stem 3 junction where 5′-end threading takes place would be expected to reduce the likelihood of threading. In the construct measured in Fig. 2, the duplex handle was separated from the end of stem 1 by a 6-nt single-stranded spacer, to minimize potential interactions with the duplex during folding. We first re-measured the FECs after reducing the length of the spacer to only 1 nt (Fig. 3a). We found the same contour length changes as before (Supplementary Table 1), but now the lower-force peak in $p(F_u)$ was increased from a small shoulder to a prominent peak (Fig. 3b), with the fraction of FECs showing full-length $\Delta L_c$ attributed to N′ doubling to 20 ± 3% of the FECs, while the fraction attributed to N decreased correspondingly to 80 ± 3%. To induce even stronger interference of the duplex handle with the pseudoknot folding, we extended the handle past the 5′ end of the pseudoknot so that it paired with the first two nucleotides in stem 1 (Fig. 3c, inset). The FECs measured using this construct (Supplementary Fig. 2) revealed an unfolding force distribution with an even greater increase in the occurrence of N′, more than doubling again to 45 ± 4% of the curves with full-length $\Delta L_c$, and a corresponding decrease in N to 55 ± 4% (Fig. 3c). We also confirmed that a 6-nt spacer was sufficiently long to avoid interference of the handle with the pseudoknot folding by measuring a construct with a 12-nt spacer, finding that the proportions of N and N′ were unchanged (Supplementary Fig. 3). Extending the handle duplex closer to the 5′ end thus produced a clear trend, suppressing N but enhancing N′ (Fig. 3d and Supplementary Table 3). However, the occupancy of Alt was effectively unchanged with spacer length (Fig. 3d, brown, and Supplementary Table 3), and the landscape parameters obtained from fitting the two populations (Fig. 3b, c, red) also remained the same within error (Supplementary Table 2). The fact that the only significant effect of changing the handles was to rebalance the N:N′ ratio supports the conclusion that N is a 5′-threaded conformer, whereas N′ is an unthreaded conformer.

Turning to the refolding FECs, we found that the pseudoknotted conformers always refolded through one or more intermediate states. The first refolding transition, which was in the force range ~10–15 pN, had $\Delta L_c = 17.9 ± 0.6$ nm (Fig. 2c), consistent with the value of 16.7 nm expected for folding of stem 1 in both the threaded and unthreaded models[18] (Supplementary Table 1 and Supplementary Fig. 4). After forming stem 1, often the rest of the pseudoknot appeared to refold all at once (Fig. 2c, red), without detectable intermediates, but other times it was seen

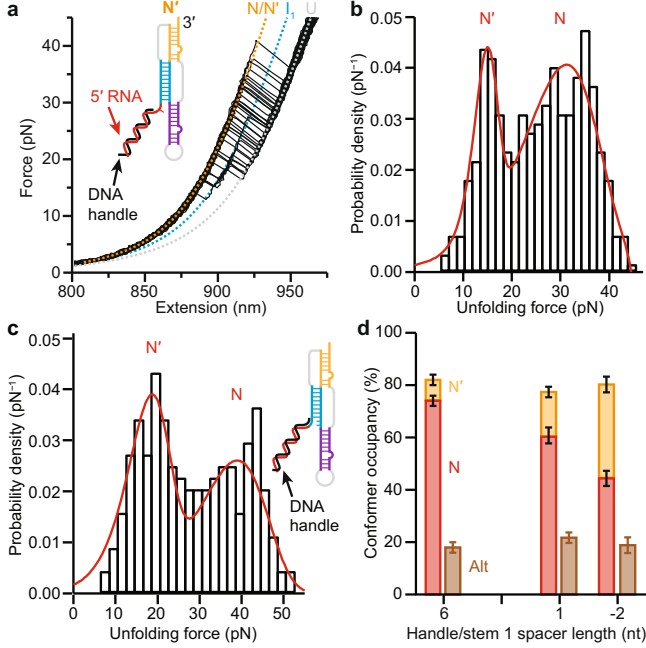

**Fig. 3 Occlusion of 5′ end suppresses folding of threaded conformer.**
**a** Unfolding FECs measured with 1-nt spacer between duplex handle and pseudoknot show same length changes as in Fig. 2. Dashed lines show WLC fits. Inset: schematic showing location of handle duplex. **b** Unfolding force distribution with 1-nt spacer between handle and pseudoknot ($N = 342$ FECs from 8 molecules) shows more prominent low-force peak and less prominent high-force peak. **c** Re-measuring unfolding with handle extending 2 nt into stem 1 of pseudoknot (inset: schematic showing location of handle duplex) yields unfolding force distribution ($N = 220$ FECs from 3 molecules) with even more prominent low-force peak and even less prominent high-force peak. **d** Fraction of FECs showing higher-force pseudoknot unfolding (red) decreases systematically as duplex handle is extended towards stem 1 of pseudoknot (or invades it), with compensating increase in fraction showing lower-force pseudoknot unfolding (orange), whereas fraction showing alternative stem-loop unfolding (brown) remains constant. Error bars on occupancies of pseudoknotted (sum of N and N′) and alternate conformers (Alt) represent standard error of proportion from numbers of FECs showing each conformation type; error bars on fraction of pseudoknotted conformers occupying N/N′ represent fitting errors from analysis of unfolding force distributions with two-population landscape model (errors correlated for N and N′, as shown). $N = 1825$ FECs from 10 molecules for 6-nt spacer, 491 FECs from 8 molecules for 1-nt spacer, and 270 FECs from 3 molecules for −2-nt spacer.

to refold through an additional intermediate (Fig. 2c, blue). The cumulative length changes from the unfolded state for these subsequent transitions were $\Delta L_c = 29.2 \pm 0.6$ nm and $35.3 \pm 0.6$ nm, consistent with expectations respectively for folding stem 3 in addition to stem 1 (29.8 nm), and then for folding stem 2 as well to form the complete pseudoknot (matching the value seen for complete unfolding). Stem 2 thus folded last, after stem 3. Intriguingly, this order is precisely what is needed to form a 5′-threaded fold topology: stem 2 must form last, after the 5′ end is threaded across the junction between stems 1 and 3. To confirm the identification of these structures in the intermediates, we repeated the measurements using anti-sense oligos to block the formation either of stem 1 (oligo 1) or stem 2 and part of stem 3 (oligo 2), as shown in Fig. 4a. The initial refolding transitions with oligo 2 present (Fig. 4b) showed effectively the same $p(F_r)$ (Fig. 4c, red) as without the oligo (Fig. 4c, black), and close to the same $\Delta L_c$, too, albeit elongated by an extra ~2 base-pairs formed with the part of stem 3 liberated by oligo 2 (Supplementary Table 1

and Supplementary Fig. 4e), confirming that stem 1 was first to refold. FECs with oligo 1 (Fig. 4d), on the other hand, showed refolding at notably lower force than stem 1 (Fig. 4c, blue); unfolding proceeded via an intermediate with a length corresponding to the lower half of stem 3, $\Delta L_c = 8 \pm 1$ nm (Supplementary Table 1 and Supplementary Fig. 4d).

We also tested the importance of $Mg^{2+}$ for the folding and stability of the pseudoknot by re-measuring FECs in the absence of $Mg^{2+}$ (Fig. 4e). We found that almost all (97%) of the curves showed the length change for pseudoknot unfolding, although $\Delta L_c$ was ~1 nm shorter than previously (Supplementary Table 1), indicating that the absence of $Mg^{2+}$ disfavored Alt. Two populations were still present in $p(F_u)$ (Fig. 4f), but the high-force population (N) was greatly reduced from what was observed with $Mg^{2+}$ present (Fig. 2d), down to only $40 \pm 8\%$ of the FECs showing full-length unfolding, and it peaked at lower forces. $Mg^{2+}$ was therefore not required for folding of the pseudoknot, but it played a key role in promoting the formation of the higher-force population attributed to the ring-knot fold topology. Fitting $p(F_u)$ to characterize changes in the landscape (Fig. 4f, red), we found that $\Delta G^{\ddagger}$ was little changed, but $\Delta x^{\ddagger}$ was significantly higher, rising to $4.0 \pm 0.6$ nm for N′ and $2.8 \pm 0.7$ nm for N (Supplementary Table 2). The pseudoknots were thus much less rigid without $Mg^{2+}$.

Finally, we examined the thermodynamic stability of N and N′ by using the Jarzynski equality[44] to estimate the free-energy change relative to the unfolded state based on the non-equilibrium work done during unfolding[45], while accounting for non-equilibrium populations of N and N′[46]. The stability of the threaded conformer N in the presence of $Mg^{2+}$ was estimated as $\Delta G_N = 61 \pm 7$ $k_B T$. This value was nominally somewhat higher than the stability of the unthreaded conformer, estimated as $\Delta G_{N'} = 55 \pm 6$ $k_B T$, but similar within the error, which was relatively large because the unfolding was not near equilibrium. Repeating the analysis for the FECs measured without $Mg^{2+}$, we found stabilities of $55 \pm 2$ $k_B T$ and $53 \pm 2$ $k_B T$, respectively, for N and N′, suggesting that the threading does not significantly change the thermodynamic stability of the pseudoknot, even though it does change the mechanical stability.

## Discussion

These results confirm the suggestion from simulations and cryo-EM imaging that the SARS-CoV-2 frameshift signal can form a variety of different structures. The state N, which was by far the most common conformation under physiological-like conditions (handle far from stimulatory structure, with $Mg^{2+}$ but without oligos), unfolded through the full length of the pseudoknot at moderately high force. This conformation was suppressed significantly by occlusion of the 5′ end by the duplex handle, precisely as would be expected for a 5′-end threaded structure such as those seen in cryo-EM images on and off the ribosome[10,15] or predicted from simulations[18]. In contrast, the conformation unfolding at lower force, N′, while occurring ten-fold less frequently than N under normal conditions, increasingly replaced N as occlusion of the 5′ end suppressed the occupancy of N, as would be expected for a conformation in which the 5′ end remains unthreaded. Unthreaded conformers have been predicted computationally[18] but not yet characterized structurally in experiments, although some individual cryo-EM images show the straight morphology expected for unthreaded conformers, in contrast to the bent shape of threaded conformers[10].

The picture of the pseudoknot folding and unfolding that emerges from this work is illustrated in Fig. 5. Stem 1 always folds first, followed by sequential folding of stem 3 and then stem 2. The orientation of the 5′ end at the moment of stem 2 formation leads to two distinct fold topologies that cannot interconvert: 5′-

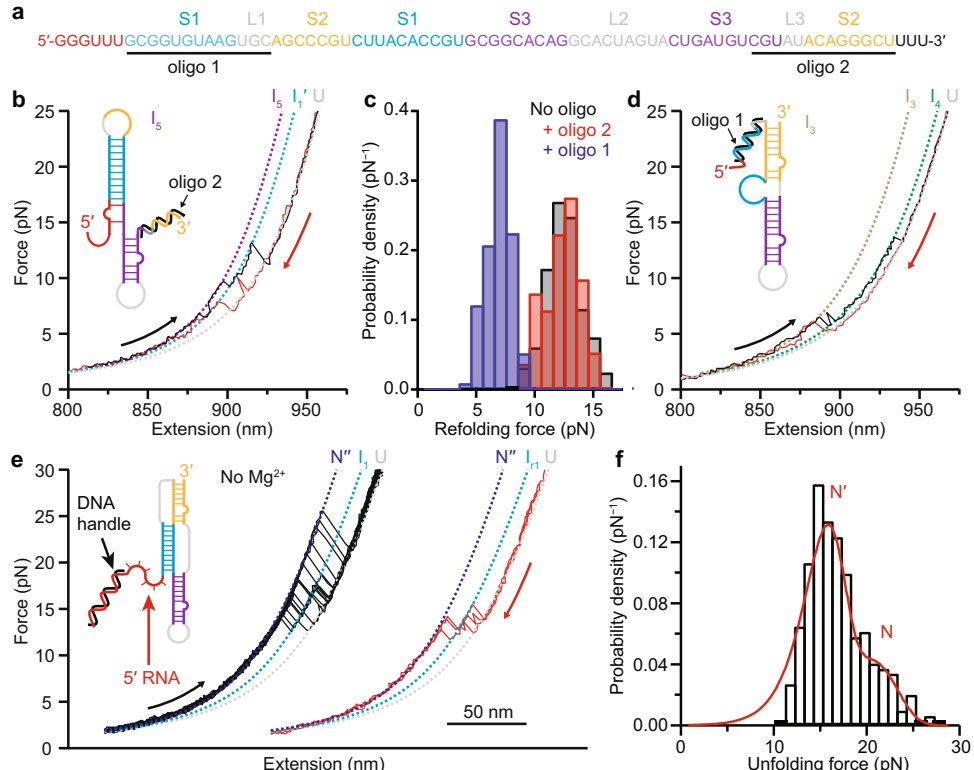

**Fig. 4 Effects of anti-sense oligomers and Mg²⁺ ions. a** Anti-sense oligo 1 blocks formation of stem 1, oligo 2 blocks formation of stem 2. Participation of bases in secondary structure indicated by the same colors and labels as in Fig. 1a. **b** FECs with oligo 2 present unfold (black) and refold (red) through an extended stem 1 (I₁'). Dashed lined: WLC fits (purple: I₅, cyan: I₁', gray: U). Inset: cartoon showing effect of oligo binding. **c** Refolding force distribution with oligo 2 present (red, N = 247 FECs from 3 molecules) is same as without oligos (black, N = 416 FECs from 10 molecules), but higher than with oligo 1 present (blue, N = 116 FECs from 2 molecules). **d** FECs with oligo 1 present (black: unfolding, red: refolding) show folding of stems 2 and 3 but not stem 1, through an intermediate (I₄) consistent with the lower part of stem 3. Dashed lines: WLC fits (brown: I₃, green: I₄). Inset: cartoon showing effect of oligo binding. **e** FECs for construct with 6-nt spacer in absence of Mg²⁺ show unfolding (black) and refolding (red) of pseudoknotted structures (N'') similar to FECs with Mg²⁺, through same intermediate (I₁ in unfolding curves, I_r1 in refolding curves), but with slightly shorter contour length changes. Curves offset for clarity. Dashed lined: WLC fits (navy: N'', cyan: I₁/I_r1). **f** Unfolding force distribution (N = 290 FECs from 2 molecules) shows two peaks, both at lower force than with Mg²⁺. Red: two-population fit to landscape model.

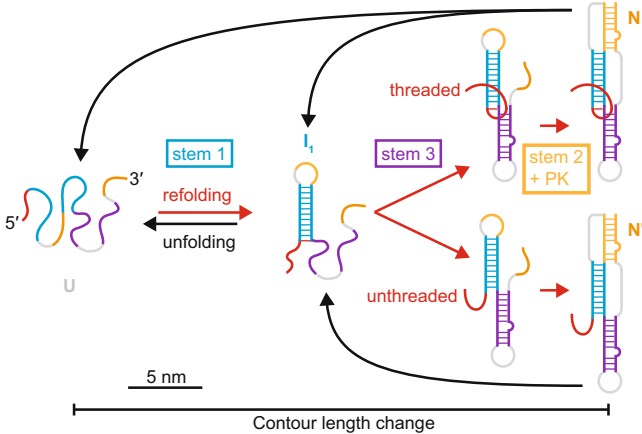

**Fig. 5 Pseudoknot folding and unfolding pathways.** Schematic of pathways for folding (red) and unfolding (black) the pseudoknotted conformers. Stem 1 forms first, then stem 3, with a bifurcation leading to 5′-threaded (top) or -unthreaded (bottom) conformers depending on orientation of 5′ end just before stem 2 forms. RNA segments are colored corresponding to their participation in native secondary structure elements as in Fig. 1a.

threaded or -unthreaded. These two topologies give rise to distinct unfolding behaviors: higher forces for the threaded fold, lower forces for the unthreaded fold. The partitioning of the folding at the point when stem 2 forms—depending on whether or not the 5′ end is lying across the stem 1/stem 3 junction, as required for threading—ensures the presence of both threaded and unthreaded conformers, with the minority unthreaded state populated at some finite level, similar to what was seen in the folding of the Zika exoribonuclease-resistant RNA (xrRNA)[21]. This folding mechanism is dependent on stem 2 folding last; as it happens, stem 2 is also predicted by mfold[47] to be the least stable thermodynamically, whereas stem 1 is expected to be the most stable, so that the folding is ordered by the relative stabilities of the stems as seen previously for two-stem pseudoknots[34,48]. Intriguingly, this order is also the same one in which the stems would refold co-translationally as the ribosome leaves the frameshift signal: stem 1 is at the 5′ end and would be expected to refold first, whereas stem 2 is at the 3′ end and would not be able to refold until after the whole pseudoknot emerged from the ribosome. Notably absent from this picture of the folding, however, is a third fold that was predicted computationally with the 3′ threaded through the stem 2/stem 3 junction[18], which makes sense given that this fold requires stem 1 to form last instead of first.

The energy landscape parameters for unfolding states N and N′ further support this picture, in particular $\Delta x^{\ddagger}$, which reports on the mechanical rigidity of the RNA structure. Smaller $\Delta x^{\ddagger}$ implies a rigid structure that is less sensitive to tension and hence more likely to rupture in a brittle manner at high force, whereas larger $\Delta x^{\ddagger}$ implies a compliant structure that is more sensitive to tension and ruptures in a lower, narrower range of forces. State N was much more mechanically rigid than N′, with $\Delta x^{\ddagger}$ roughly three times smaller, which makes sense in terms of the mechanical effects of threading: threading of the 5′ end should rigidify the fold via interactions between the 5′ end and the helical junction in the pseudoknot[18] that constrain the motion of the terminus in response to tension applied to the 5′ end, compared to the unthreaded fold. Indeed, the $\Delta x^{\ddagger}$ value of only 0.7 nm for state N is among the smallest reported for any pseudoknot, signifying the particular rigidity of this unusual fold topology; in contrast, the $\Delta x^{\ddagger}$ value for N′ is comparable to the range of values more typically reported for other pseudoknots, ~1.5–2 nm[22,40].

Even though the observation of state N in the absence of $Mg^{2+}$ shows that $Mg^{2+}$ is not absolutely required for the pseudoknot folding, consistent with results from NMR[16], $Mg^{2+}$ clearly plays an important role. Given that $\Delta x^{\ddagger}$ increased significantly for both N and N′ in the absence of $Mg^{2+}$, the latter must be essential for stabilizing tertiary contacts that rigidify both the threaded and unthreaded folds. Structural studies have not yet resolved coordinated $Mg^{2+}$ ions or determined how $Mg^{2+}$ affects the pseudoknot structure. However, comparing NMR results to computational modeling[16,18] suggests that $Mg^{2+}$-mediated tertiary interactions may be especially important in the stem 2/loop 1 region, possibly explaining why both N and N′ are more compliant in the absence of $Mg^{2+}$. $Mg^{2+}$ must also be important in stabilizing threading of the 5′ end into the stem 1/stem 3 junction, since removing $Mg^{2+}$ greatly reduced the incidence of the threaded conformer. Presumably, $Mg^{2+}$ binds near the stem 1/stem 3 junction to help coordinate the threading, consistent with suggestions from simulations showing $Mg^{2+}$-mediated interactions between the 5′ end and stem 1 that make the three-helix junction more compact[18]. Such contacts with the threaded end in N might explain why the rigidification is twice as large for N as it is for N′ (four-fold decrease in $\Delta x^{\ddagger}$ instead of two-fold).

Comparing the SARS-CoV-2 pseudoknot to the Zika virus xrRNA, the only other RNA forming a similar ring-knot whose folding has been studied, reveals some interesting differences. The ring-knot in the Zika virus xrRNA unfolds at a much higher force, above 60 pN, acting as a mechanical road-block to digestion of the viral RNA by host exoribonucleases[21]. Such extreme mechanical resistance would be functionally counterproductive for the SARS-CoV-2 pseudoknot, however, because although the latter acts to induce a frameshift, it nevertheless must not prevent ribosomal translocation. Concurrently, these two RNAs differ in the importance of $Mg^{2+}$ for threading of the 5′ end before closure of the pseudoknot: the absence of $Mg^{2+}$ abolishes threading entirely for the Zika virus xrRNA, leading to the formation of different structures, but it has a much less dramatic effect on the SARS-CoV-2 pseudoknot, partially inhibiting rather than abolishing the threading so as to rebalance the proportions of N and N′. We speculate that the reduced role of $Mg^{2+}$ in 5′-threading in the SARS-CoV-2 pseudoknot may help to reduce the mechanical stability of the ring-knot sufficiently to allow the ribosome to unfold it during −1 PRF.

Turning to the non-pseudoknotted conformer, Alt, the FECs provide several clues to its identity. Its low unfolding force and the relatively large distance to the barrier found from fitting the force distribution (Supplementary Fig. 1, red), $\Delta x^{\ddagger} = 4.5 \pm 0.8$

nm, indicate that it involves secondary structure only. Moreover, the length change reveals that 46 ± 2 nts are folded in Alt, and the fact that Alt does not rapidly convert into N or N′ implies that it involves stems that differ from those in the pseudoknotted conformers. The most likely structure consistent with these results is the hairpin with multiple bulges shown in Supplementary Fig. 4c. Because Alt is much less stable than N/N′, it would be expected eventually to convert into a pseudoknot, given enough time. Indeed, analyzing only the first FEC measured for each molecule (for which the RNA had much more time to find the minimum-energy state than during repeated unfolding/refolding cycles) supports this picture: the fraction of FECs starting in Alt was reduced significantly, by roughly a factor of 3, to 6 ± 4%. Moreover, this conversion was occasionally seen directly, in rare examples where the RNA folded into Alt but converted to N/N′ during the subsequent unfolding FEC (Supplementary Fig. 5). Alt was almost eliminated in the absence of $Mg^{2+}$, suggesting that $Mg^{2+}$ stabilizes it and helps to trap the RNA in Alt kinetically[49].

The significant heterogeneity seen here for the SARS-CoV-2 frameshift signal is entirely consistent with the direct correlation between conformational heterogeneity and −1 PRF efficiency found in recent work[27], given the relatively high level of −1 PRF in SARS-CoV-2 observed in functional assays[1,11], underscoring the functional relevance of understanding the force-dependent conformational dynamics. We note that the SMFS assay mimics several physiologically important features of −1 PRF: the stimulatory structure is indeed under tension applied directly by the ribosome[28], this tension is ramped up and down as the ribosome attempts to unfold the RNA during −1 PRF[30], and the stimulatory structure undergoes repeated unfolding/refolding cycles as multiple ribosomes translocate through it, sometimes in rapid bursts[50]. However, SMFS does not perfectly recapitulate the circumstances in the cell. For example, the ribosome only applies force to the 5′ end of the RNA (not both ends as in the tweezers), and the force profile over time is more complex, including periods of sustained tension on the RNA while the ribosome is paused at the frameshift site[28], in addition to ramps up and down. Specific contacts between the SARS-CoV-2 pseudoknot and the ribosome are also proposed to play a role in −1 PRF[15]. As a result, the forces needed to unfold the RNA in SMFS may differ from those involved in unfolding it in the cell. Although the duplex handles in the SMFS assay are not present in the cell, the fact that no change is seen when moving the handle from 6 to 12 nt away from the pseudoknot suggests that a 6-nt spacer is sufficient for the handle to have little to no effect on the folding.

Finally, we note that the existence of distinct fold topologies has important implications for structure-based drug-discovery efforts targeting the SARS-CoV-2 frameshift signal, because the structure of the junction between the helices in the pseudoknot, which is the locus of the most likely binding pockets for small molecules[14,51], is strongly affected by whether the 5′ end is threaded or not[18]. Combining this observation with the previous result showing that −1 PRF efficiency varies linearly with the conformational heterogeneity as measured by the Shannon entropy[27] suggests a strategy for developing small-molecule modulators of −1 PRF in SARS-CoV-2: ligands that stabilize the 5′-threaded conformer (thereby decreasing the heterogeneity) should be sought for inhibiting −1 PRF, whereas ligands that stabilize the unthreaded conformer (thereby increasing the heterogeneity) should be sought for enhancing −1 PRF. Future work characterizing the interactions of the SARS-CoV-2 frameshift signal with ligands that have been found to modulate −1 PRF should help clarify the molecular mechanisms of action.

## Methods

**Sample preparation**. Samples consisting of a single RNA strand linked at each end to double-stranded handles were prepared in two ways. (1) An RNA strand containing the SARS-CoV-2 pseudoknot and spacer sequences (Fig. 1a and Supplementary Table 4) flanked by long handle sequences was annealed to single-stranded (ss) DNA complementary to the handles sequences[22]. The DNA fragment corresponding to the sequence in Fig. 1a was cloned into the pMLuc-1 plasmid between the BamHI and SpeI sites. A 2749-bp DNA transcription template was amplified by PCR from this plasmid, containing a T7 promoter in the upstream primer, followed by a 1882-bp handle sequence, the pseudoknot in the middle, and then a 798-bp handle sequence downstream; RNA was transcribed from this template in vitro. Two ssDNA handles complementary to the upstream and downstream handle sequences in the RNA were created by asymmetric PCR. The 3′ end of the 1882-nt DNA handle was labeled with dig-ddUTP using terminal transferase (Roche), and the 798-nt DNA handle of the transcript was functionalized with biotin on the 5′ end of the PCR primer. The RNA transcript was then annealed to the DNA handles, completing the construct. (2) In the second method, a shorter RNA strand was annealed to a 300-nt ssDNA handle on one end, and to the overhang on a 2094-bp double-stranded (ds) DNA handle on the other end. The dsDNA handle, labeled with digoxigenin via the upstream primer, was prepared by digesting a 2075-bp PCR product amplified from the plasmid pUC19 with PspGI, and then ligating a 56-nt DNA oligo to the digest sticky end to create a 36-nt 3′ overhang. RNA was transcribed in vitro from a 406-bp DNA template containing a T7 promoter, the 36-nt sequence complementary to the ssDNA overhang, the pseudoknot and spacer sequences (Fig. 1a), and a 300-nt handle sequence; this transcription template was made by cloning the required DNA sequences into a modified pMLuc-1 vector between the XhoI and SpeI sites. A 300-nt ssDNA handle complementary to the handle region of the RNA transcript was made by asymmetric PCR and annealed to the RNA, as above. The resulting DNA-RNA complex was then annealed to the overhang of the dsDNA handle, completing the construct. The sequences of the primers used in the two construct designs are listed in Supplementary Table 5.

The RNA/handle constructs were readied for measurement by diluting them to ~160 pM, mixing them with equal volumes of 600- and 820-nm diameter polystyrene beads (coated respectively with avidin DN and anti-digoxigenin) at concentrations of ~250 pM, and incubating the mixture for ~1 h at room temperature to create RNA-bead dumbbells. The incubation was then diluted ~100-fold into RNase-free measuring buffer: 50 mM MOPS pH 7.5, 130 mM KCl, 4 mM MgCl₂, and 200 U/mL RNase inhibitor (SUPERase•In, Ambion). An oxygen scavenging system consisting of 40 U/mL glucose oxidase, 185 U/mL catalase, and 250 mM D-glucose was also included in the buffer. The diluted dumbbells were placed in a flow chamber prepared on a microscope slide and inserted into the optical trap.

**FEC measurements and data analysis**. Measurements were made using custom-built optical traps described previously[38] controlled by Labview 2018.0.1. Traps were calibrated for position detection for each dumbbell prior to measurement following standard methods[52]. FECs were measured by moving the traps apart at a constant speed of ~160 nm/s to increase the force up to ~50 pN and unfold the RNA, bringing them back together at the same speed to ramp the force back down to ~0 pN, waiting 5–10 s to allow refolding, and then repeating the cycle. Trap stiffnesses were 0.45–0.62 pN/nm. Data were sampled at 20 kHz and filtered online at the Nyquist frequency. Measurements with anti-sense oligos added oligo 1 or oligo 2 to the measuring buffer at a final concentration of 10 μM. For measurements in the absence of $Mg^{2+}$, MgCl₂ was removed from the measuring buffer and EDTA was added to a final concentration of 1 mM.

Each of the branches of the FECs separated by "rips" representing unfolding/refolding transitions was fit to an extensible WLC model relating the applied force, $F$, and molecular extension, $x$:

$$F(x) = \frac{k_B T}{L_p} \left[ \frac{1}{4} \left( 1 - \frac{x}{L_c} + \frac{F}{K} \right)^{-2} - \frac{1}{4} + \frac{x}{L_c} - \frac{F}{K} \right], \quad (1)$$

where $L_p$ is the persistence length, $L_c$ the contour length, and $K$ the enthalpic elasticity[53]. Two WLCs in series were used for the fitting, one to describe the duplex handles and the other for the unfolded RNA[38]. The WLC parameters for the handles, found from fitting the folded state of the FECs, were typically $L_p$ ~ 40 nm, $L_c$ ~ 850 nm for the shorter construct and ~950 nm for the longer one, and $K$ ~ 1000 pN. The parameters for the unfolded RNA were fixed at $L_p = 1$ nm, $L_c = 0.59$ nm/nt, and $K = 1500$ pN, so that the only free parameter was the number of nucleotides unfolded[38].

Force distributions were fit to the theory of Dudko et al.[35], using

$$p(F) \propto \frac{k(F)}{r} \exp \left\{ \frac{k_0}{\beta \Delta x^{\ddagger} r} - \frac{k(F)}{\beta \Delta x^{\ddagger} r} \left( 1 - \frac{2 \Delta x^{\ddagger} F}{3 \Delta G^{\ddagger}} \right)^{-1/2} \right\}, \quad (2)$$

where $k(F) = k_0 \left( 1 - \frac{2 \Delta x^{\ddagger} F}{3 \Delta G^{\ddagger}} \right)^{1/2} \exp \left\{ \beta \Delta G^{\ddagger} \left[ 1 - \left( 1 - \frac{2 \Delta x^{\ddagger} F}{3 \Delta G^{\ddagger}} \right)^{3/2} \right] \right\}$, $k_0$ is the unfolding rate at zero force, $\Delta x^{\ddagger}$ is the distance from the folded state to the barrier,

$\Delta G^{\ddagger}$ is the barrier height, and $1/\beta = k_B T$ is the thermal energy. Distributions of the unfolding forces were fit by a sum of two independent distributions defined by Eq. 2, representing the contributions from unfolding the independent and non-interconverting states N and N′. For the distributions measured from the constructs with 6- and 12-nt single-stranded spacers, where two peaks in the distribution were not immediately obvious, we verified that two-population fits of $p(F_u)$ were justified using the AIC[39] to judge the relative likelihood of the one- and two-population models and the Wald–Wolfowitz runs test to assess the randomness of the fit residuals. These tests rejected the single-population fits in favor of the double-population fits in each case: the difference in AIC values was 11.1 with the 6-nt spacer (0.4% likelihood one-population model is better) and 9.3 with the 12-nt spacer (0.9% likelihood one-population model is better); the Wald–Wolfowitz test statistic values were 1.22 (6-nt spacer) and 1.81 (12-nt spacer), both lower than the critical value of 1.96 (95% confidence level). Errors for the fitting parameters were found from bootstrapping analysis.

The thermodynamic stabilities of N and N′ were determined by calculating distributions of work done during unfolding, found by integrating the fitted unfolding FECs from the extension corresponding to $F = 2$ pN up to the point where the unfolded state was reached, while subtracting the work done to stretch out the unfolded RNA (found from integrating the WLC for the unfolded state between the same two end-points)[42]. Because N and N′ have very similar $\Delta L_c$ and their unfolding force distributions overlap significantly, it was not possible to assign definitively the initial state of any given FEC to N or N′, as needed to build the work distribution for each state. Instead, we used a probabilistic approach, determining the relative likelihood that a given FEC unfolded from N or N′ based on its unfolding force, using the fits of the unfolding force distributions to Eq. 2: the likelihood that a curve with unfolding force $F_0$ started in state N (or N′) was given by $p_{N(\text{or } N')}(F_0)/[p_N(F_0) + p_{N'}(F_0)]$. We used this likelihood function to assign each curve to N or N′ while sampling a number of curves equal to the total number measured, with replacement, thereby generating the unfolding work distribution for each state for this sampling. We then calculated the free energy for unfolding from the Jarzynski equality[44], $\Delta G = -k_B T \ln[\langle \exp(-W/k_B T) \rangle]$, where $W$ is the work done. We corrected for the bias in the Jarzynski estimate[54], using the weighted sum of the free energy values to calculate the dissipated work. We also corrected for the non-equilibrium populations of N and N′[46], adding an additional energy $-k_B T \ln (1/\phi)$, where $\phi$ is the fraction of refolding curves that end in the state (N or N′) whose stability is being calculated. This procedure was then repeated 5000 times while resampling the curves and recalculating their assignments to N or N′, yielding the average values and standard deviation for $\Delta G$ reported for N and N′. All analysis was done using Igor Pro 7.08.

**Reporting summary**. Further information on research design is available in the Nature Research Reporting Summary linked to this article.

## Data availability

The data supporting the findings of this study are available from the corresponding authors upon reasonable request. The raw data of this work[55] have been deposited in Figshare (https://doi.org/10.6084/m9.figshare.14614176).

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

## Acknowledgements
This work was supported by the Canadian Institutes of Health Research grant reference number OV3-170709, Alberta Innovates, and National Research Council Canada.

## Author contributions
K.N., M.Z., and M.T.W. designed the research; K.N., S.M., D.B.R., and S.M.I. provided reagents; K.N., M.Z., A.L., N.Q.H., and A.N. performed experiments; K.N., M.Z., A.L., and M.T.W. analyzed data; K.N., M.Z., and M.T.W. wrote the manuscript; and all authors edited the manuscript.

## Competing interests
The authors declare no competing interests.
