## [Peer Review File · Nature Communications]

REVIEWER COMMENTS

Reviewer #1 (Remarks to the Author):

The present work by Neupane et al studies the diversity and conformational mechanics of RNA pseudoknot molecules from SARS-CoV-2. They use an optical trap setup with the usual DNA handle design. They found that the mechanical stability measured can provide information regarding the conformation of the RNA, threaded vs unthreaded knot which is important for the -1 frameshift. This type of work is important as it reveals that mechanical force offers information during molecular processes related to viral infections. However, I found several issues that the authors should address:

1. I found that the issue with the handle is confusing and perhaps unnecessary for the story. In figure 3, they show how the handle occlusion in the 5' end affects the mechanical behavior of the RNA molecule. To me this is simply an artifact. In fact, when they move the handle 6 nt, the low-force peak is less prominent. The author should only concentrate on the results that are not affected by the artificial handle addition and provide a biological context to them. They should move the extra info to the supplementary as a test. I am not suggesting this data should not be part of the work, I only think it creates a confusion between what is technically relevant versus what is biologically relevant. Why do the authors think that occlusion of 5' increases the high-force peak proportion?

2. Since the handles obviously have an influence in the balance N/N' . What would be the conformers balance in the absence of handles? What would be the difference in conformers proportion if the authors use the same molecule in repeating cycles versus using different molecules in each unfolding trajectory?

3. How does the data correlate with forces applied in the ribosome? According to previous studies these forces are generally below 20 pN which means that they high-force peak might not even be reached in the ribosome. Could that be the case? The authors should discuss their results in light of the previous findings regarding molecular mechanics in the ribosome.

4. One of the most interesting points of this work is the possibility of using small-molecule modulators to prevent the -1 frameshift. However this is just mentioned. The concept might have some important implications. Since it seems that there are already screening tools for such drugs, have the authors considered trying such molecules?

Reviewer #2 (Remarks to the Author):

Neupane et.al use optical tweezers to generate force extension curves for the SARS2 frameshift stimulation element (FSE) and investigate the folding pathway as well as find alternative conformations for this RNA. The key findings are evidence that 1) the FSE forms threaded and unthreaded structures and at least one alternative structure and 2) the unfolding pathway is $S2/S3 \diamond S1$ and the folding pathway is $S1/S3 \diamond S2$ consistent with threading. The authors are experts in optical tweezers and the data is clear, well- presented and analyzed. This work is convincing as an in vitro biophysical study. However, there is little functional or physiological relevance of the authors findings and the interest to a wider field is low.

The threaded form has been seen before by cryoEM (R.Das lab, bioRxiv). The functional significance for these alternative structures are entirely based on speculations. The biggest concern is the physiological relevance -1. the FSE is taken out of its native context 2. DNA handles are placed and 3. Unfolding and refolding by optical tweezers doesn't recapitulate unfolding by ribosome – the ribosome has a helicase and goes in sequentially along the RNA.

The authors clearly show that the DNA handle (which is not physiological) affects the structures.

Overall, this is an interesting and well performed biophysical study of a short RNA piece from SARS-Cov2 but because the physiological relevance of the findings is entirely unclear I would not recommend for publication in Nature Communications.

Reviewer #3 (Remarks to the Author):

In this paper, Woodside and collaborators investigate the RNA pseudoknot of Sars-Cov-2 virus using force spectroscopy finding two distinct folded structures (N and N') with different kinetic stabilities. N and N' cooperatively unfold in a single rip where the approximately 70 nucleotides of the RNA release an extension of 35nm. The two structures can be discriminated from the bimodal rupture force distributions that show that N has larger kinetic stability and brittleness as compared to N'. The authors claim that N,N' correspond to the threaded and unthreaded conformations of the RNA pseudoknot depending on which side the 5' end of the RNA passes through the stem1/stem3 junction. To support this hypothesis additional experiments are carried out by extending the hybrid RNA/DNA 5' handle to occlude the junction, using antisense oligos to inhibit stems 1 and 3, and test the results without magnesium. In my view the conclusions are well supported by the findings, the results are well described and the text and figures clear. Overall this is a beautiful and elegant paper that I am happy to recommend for publication after the authors have considered the following (general) comments.

In the intro, page 3, four lines before the Results section. "Stem 1 usually folds first, followed by stem 3 and lastly stem 2" I think it should be the other way around, " ...Is there any energetic or kinetic explanation for this preferred S1-S2-S3 folding pathway?

2. The focuses on structures N and N', but what can be said from the other 20% second class of alternative structures? Are there any predictions for these? Does the 20% of alternative structures change in the absence of magnesium? Are these misfolded structures. In such a case I would expect a reduction of this 20% fraction in the absence of magnesium (as magnesium specific binding stabilizes native by rescuing misfolded structures).

3. In the schematics of the RNA structure in Figure 2B it is not clear if this is the threaded or unthreaded structure. I presume it can be both but this should be said in the caption. Then In the black pulling curves in the leftmost set, there are also curves showing the intermediate I1. Is this the same selected set of magenta curves that are shown in the middle set of FEC?

4. The part on experiments without magnesium is interesting because it shows that, although magnesium further stabilizes N with respect N', it is not essential for the existence of N. Could the authors say something about the specific magnesium contacts in the stem1-stem 3 junction? Has the structure of the RNA pseudoknot been resolved (e.g. with cryo-X ray diffraction or NMR) and the magnesium binding sites been determined? What are the conformational changes induced by magnesium specific binding? Do these conformational changes occur close to the stem1-stem 3 junction? In the conclusions the authors discuss differences with the Zika virus xRNA where threading is abolished in the absence of magnesium. How the conformational changes induced by magnesium in the RNA pseudoknot of the Zika virus differ from Sars-Cov-2?

5. The authors analyze the different kinetic stabilities of N and N' by calculating the distance to the transition state x^{++} and the kinetic barrier DG^{++} . However, it would be interesting to say something about the thermodynamic stabilities of the two states. Although N is kinetically more stable than N', it is reasonable to expect similar thermodynamic stabilities for the threaded and unthreaded conformations. In my view, the different topologies of the threaded and unthreaded conformations would mostly change the conformational entropy of the 5' end and this difference should not amount much to the free energy. Moreover a similar thermodynamic stability N vs N' would be also consistent

with a regulatory feature of the PRF of the RNA pseudoknot. Could the authors derive equilibrium free energies from their data? I am aware that the FEC are irreversible but there are powerful methods to extract the free energies in pulling experiments based on classifying (from the rupture force histograms, Fig. 2D) the unfolding trajectories into N and N' states and using a fluctuation theorem for kinetic states (e.g. Alemany et al., Nat. Phys. 8 (2012) 688-694).

6. I recommend calling figure 5 while showing and discussing figure 2 to better understand the experimental results of Figure 2 and better understand the threading vs non-threading differences.

Response to reviewer comments

Reviewer #1:

The present work by Neupane et al studies the diversity and conformational mechanics of RNA pseudoknot molecules from SARS-CoV-2. They use an optical trap setup with the usual DNA handle design. They found that the mechanical stability measured can provide information regarding the conformation of the RNA, threaded vs unthreaded knot which is important for the -1 frameshift. This type of work is important as it reveals that mechanical force offers information during molecular processes related to viral infections. However, I found several issues that the authors should address:

1. I found that the issue with the handle is confusing and perhaps unnecessary for the story. In figure 3, they show how the handle occlusion in the 5' end affects the mechanical behavior of the RNA molecule. To me this is simply an artifact. In fact, when they move the handle 6 nt, the low-force peak is less prominent. The author should only concentrate on the results that are not affected by the artificial handle addition and provide a biological context to them. They should move the extra info to the supplementary as a test. I am not suggesting this data should not be part of the work, I only think it creates a confusion between what is technically relevant versus what is biologically relevant.

We regret that the way the results with different handle locations were presented was confusing. The purpose of including data from measurements where the handle occluded the 5' end to different extents was to demonstrate that the two populations, N and N', reflect threaded and unthreaded conformers of the pseudoknot, respectively. We had thought that presenting the results from the intermediate case first (handle located 1 nt away from the pseudoknot), where the two populations could be very readily observed, and then later showing the results from the more native-like condition (handle located further away) where the presence of two populations is less obvious, would make the results easier to follow. However, clearly that was not the case! We have revised the manuscript in the Results section (Figs 2 and 3) to change the order in which we present the results, showing first the results from the measurements with the handle 6 nt away from the pseudoknot (far enough away not to affect the folding, as discussed below in response to point 2), since these are the ones that are most relevant biologically in terms of the relative prevalence of the two states. We then present the results with the handle closer to the RNA, such that it perturbs the folding.

We respectfully disagree with the suggestion that the results from the other handle lengths should be moved to the SI: they are key evidence supporting the conclusion that the two populations represent threaded and unthreaded conformers (because of their sensitivity to occlusion of the 5' end), and hence should remain in the main text. Furthermore, we note that the unthreaded conformer is not a technical artifact, as there is evidence in the full ensemble of cryo-EM images of the pseudoknot (Zhang *et al. bioRxiv*) for conformers that involve linear alignment of the helices like what is expected for the unthreaded conformer, not just for the “bent” shapes of the threaded conformer. We have revised the manuscript to discuss this issue (Discussion, p. 9).

Why do the authors think that occlusion of 5' increases the high-force peak proportion?

As we show in the manuscript, occlusion of the 5' end actually **decreases** the prevalence of the higher-force state, N, the opposite of what the reviewer states here. The fact that the higher-force

state is less common when the 5' end is occluded makes excellent sense in terms of our interpretation that N has a threaded 5' end: when forming N, the 5' end is threaded into the cleft between S1 and S3 before the ring is closed by formation of S2, but if the duplex handle is extended to be very close to the 5' end, the duplex has trouble threading into this cleft because of steric hindrance from the bulky duplex, and hence the ring tends to close with the 5' end still unthreaded more commonly than it would otherwise. This steric hindrance from the duplex that disfavors threading when the handle occludes the 5' end is modest when the handle is 1 nt away from the pseudoknot, and stronger when the handle is extended to incorporate part of the 5' end of S1 into the duplex, explaining why the effect is larger in the latter case. We have revised the manuscript to explain this point more clearly (Results, p. 6).

2. Since the handles obviously have an influence in the balance N/N'. What would be the conformers balance in the absence of handles?

We note that in the context of the virus, the pseudoknot is connected to ~13.5 kb of RNA on the 5' end and ~16.5 kb of RNA on the 3' end. These segments of RNA “attached” to the pseudoknot can be viewed as similar to the handle segments of the RNA construct that we measure (although many times longer than our handles). The main difference between the full virus and our construct is that we have annealed kb-long single-stranded (ss) DNA to the RNA on either side of the pseudoknot. Hence we feel that the relevant question is not so much what happens without any “handles” at all, but rather what happens when the duplex part of the handle generated by annealing the ssDNA is moved sufficiently far away from the pseudoknot that its influence is not felt. (Note that the ssDNA is required to do the measurement, as otherwise the long RNA handles would form a plethora of structures in which we are uninterested, whose unfolding would be impossible to disentangle from the unfolding of the pseudoknot we want to study.)

We had assumed that a 6-nt single-stranded spacer from the pseudoknot (a typical length for the spacer between the pseudoknot and the slippery sequence) should be sufficient to minimize interactions between the duplex handle and the pseudoknot, but it is possible that the handle might still have some influence on the partitioning between N and N' even at that distance. We therefore prepared a new construct with a spacer with twice the length (12 nt, placing the edge of the DNA handle 7.1 nm away from the pseudoknot in contour length) and re-measured the unfolding. We found that the unfolding force distribution again showed two peaks, with the proportion of the lower-force population (N') the same within error as what was seen for the 6-nt spacer. Hence moving the DNA handle more than 6 nt from the pseudoknot does not change the results for the proportions of N and N', suggesting that the results from the construct with a 6-nt spacer give a reliable indication of what would be seen in the absence of any duplex handle.

We have revised the manuscript to include the new measurements (Results p. 6, Fig. S3).

What would be the difference in conformers proportion if the authors use the same molecule in repeating cycles versus using different molecules in each unfolding trajectory?

One might think that we could, in principle, address this question by comparing the results from the first pull measured on each molecule to the overall distributions we observed from repeated unfolding/refolding cycles. However, this analysis is problematic because it is generally not possible to be certain if the pseudoknot starts off in the N or N' conformer in any given FEC: these two conformers have very similar ΔL_c values, hence they can't be distinguished by their length changes, and they also have significant overlap in their unfolding force distributions, such

that only FECs that unfold at high forces (over ~ 30 pN) can be unambiguously assigned to a specific conformer (in this case, N). As a result, we can't calculate a simple fraction of N vs N' from the "first pulls" for the molecules measured so far—we would need to have enough measurements on different molecules to build a full distribution as in Fig. 1d, which would require measuring roughly 20–50 times as many molecules as we have done to date, which is not practical experimentally.

Although we can't properly analyze how the N/N' ratio differs for the first pulls from the rest of the measurements, we can do so for Alt vs N/N', because these states can be distinguished by their lengths. We find that out of the 31 "first pulls" from all molecules measured with Mg^{2+} but without any anti-sense oligos, only 2 started in Alt, translating into an occurrence of $6 \pm 4\%$. This occurrence is distinctly lower than seen in the cycled data (where the overall rate of Alt is $\sim 20\%$), suggesting that the metastable Alt state reconfigures into N and/or N' over moderate time frames, as one might expect given that the latter are much more thermodynamically stable.

As a final point in response to this question, we note that under physiological conditions, the pseudoknot will be constantly unfolded and refolded as ribosomes translocate along the mRNA. Hence our measurement protocol of repeated cycles of unfolding and refolding is conceptually relevant to what occurs in the cell. Indeed, measurements of translation and frameshifting on single HIV-1 mRNAs in live cells showed evidence of ribosomal "traffic jams" at frameshifting sites (Lyon *et al.* (2019) *Mol. Cell* 75:172–183), where numerous ribosomes pile up behind the frameshift signal. This work suggests that the amount of time available for the stimulatory structure to refold varies from the order of ~ 1 min (the average time between ribosome passage) to the order of a few seconds (in the case of ribosomal traffic jams), placing our ~ 5 -s refolding time within the range that is physiologically relevant.

We have revised the manuscript to discuss this question and describe the analysis of the first pulls (Discussion, p. 12).

3. How does the data correlate with forces applied in the ribosome? According to previous studies these forces are generally below 20 pN which means that they high-force peak might not even be reached in the ribosome. Could that be the case? The authors should discuss their results in light of the previous findings regarding molecular mechanics in the ribosome.

It is important to keep in mind that we are not applying force to mirror exactly what the ribosome does. Although the ribosome does actively apply force when unwinding RNA structures (Ref. 28), in the context of programmed frameshifting it appears to do so in quite a complex way. Studies of frameshifting in single ribosome molecules show that not only do ribosomes stall/pause at the stimulatory structure before it is eventually unfolded, but they can also shuttle back and forth by a few nucleotides while making multiple attempts at unfolding (Ref. 30). Hence the pseudoknot likely experiences a complex force profile, with periods where forces fluctuate up and down as well as periods of more constant tension. Unfortunately, these force profiles have never been measured reliably, and they presumably contain significant elements of randomness, so it is not really possible to replicate them in our experiments. Indeed, because the effects of mechanical force depend strongly on the loading rate (the rate at which the tension changes), it's not even very meaningful to make simple quantitative comparisons of the forces observed in constant pulling speed experiments (as in our work) to what would occur during unfolding by the ribosome (which seems to involve a combination of fluctuating as well as roughly constant force).

For example, unfolding forces for many pseudoknots have been measured from force spectroscopy studies in the 40–50 pN range (*e.g.* Refs. 22, 27, and 40 in the revised manuscript), which is above the typical maximum force applied by the ribosome (as the reviewer notes) and hence might suggest that they can't be unfolded by the ribosome. Yet every one of these pseudoknots is reliably unfolded by ribosomes during programmed frameshifting! The example of the frameshift-stimulatory pseudoknots from human endogenous retrovirus and mouse mammary tumor virus, which have unfolding forces that differ by a factor of 2 but are both unfolded reliably by the ribosome and indeed even stimulate the same level of frameshifting, shows that no simple correlations between unfolding forces in SMFS assays and the ability of the ribosome to unfold a pseudoknot can be drawn, as discussed in Ref. 22. Hence it's not correct to assume that the ribosome would be unable to unfold N because its average unfolding force in our measurements is above 20 pN.

We have revised the manuscript (Discussion, p. 13) to discuss this issue and mention that the forces needed to unfold the pseudoknot in SFMS assays may differ from the forces involved in unfolding the pseudoknot by the ribosome.

4. One of the most interesting points of this work is the possibility of using small-molecule modulators to prevent the -1 frameshift. However this is just mentioned. The concept might have some important implications. Since it seems that there are already screening tools for such drugs, have the authors considered trying such molecules?

We agree with the reviewer that this possibility is very exciting, and we are planning to do such measurements in the future. However, such work represents a major research effort in its own right that is very time-consuming, and it is well beyond the scope of the current manuscript—indeed, it requires first that we understand the folding dynamics of the pseudoknot in the absence of any compounds, which is the purpose of this manuscript. We mentioned the possibility of studying the effects of small-molecule inhibitors in the context of future possibilities, not least in the hope that other researchers who are engaged in efforts to find effective inhibitors will be stimulated to join us in efforts to understand their mechanism of action.

Reviewer #2:

Neupane et al use optical tweezers to generate force extension curves for the SARS2 frameshift stimulation element (FSE) and investigate the folding pathway as well as find alternative conformations for this RNA. The key findings are evidence that 1) the FSE forms threaded and unthreaded structures and at least one alternative structure and 2) the unfolding pathway is S2/S3 \leftrightarrow S1 and the folding pathway is S1/S3 \leftrightarrow S2 consistent with threading. The authors are experts in optical tweezers and the data is clear, well- presented and analyzed. This work is convincing as an in vitro biophysical study. However, there is little functional or physiological relevance of the authors findings and the interest to a wider field is low.

We respectfully disagree that there is little functional or physiological relevance to understanding the folding dynamics of the pseudoknot that stimulates programmed frameshifting. It has been amply demonstrated in the literature that pseudoknots do not function simply as static structures in -1 PRF—rather, their dynamic properties are an essential component of the mechanism

triggering -1 PRF (see, for example, Refs. 22–27 in the manuscript). Furthermore, as mentioned above in response to point 2 of reviewer 1, pseudoknots are constantly being unfolded and refolded mechanically as ribosomes translocate through them while translating the viral mRNA. The repeated unfolding-refolding cycles that result from the passage of the ribosomes provide the RNA multiple opportunities to explore alternative possibilities for the stimulatory structure upon refolding, reinforcing the physiological relevance of understanding the dynamic properties of stimulatory structures and the alternative folds they may take on. Our quantitative characterization of the heterogeneous structural dynamics of this pseudoknot therefore has both functional and physiological relevance to -1 PRF in SARS-CoV-2.

Moreover, our work demonstrates a number of important advances that should be of interest to a wide audience:

- Characterizing for the first time the structural dynamics of the frameshifting pseudoknot from SARS-CoV-2—a feature of the virus that is important functionally to the virus and hence a possible drug target—and doing so under tension, the condition relevant for frameshifting.
- Demonstrating empirically that this pseudoknot forms not only different conformers, but different fold topologies, something not seen previously in frameshift-stimulatory pseudoknots that has relevance for targeting frameshifting therapeutically.
- Determining for the first time the folding mechanism of a frameshift signal with a knot-like structure—a fold topology not previously studied in frameshift-stimulatory structures.
- Revealing how Mg^{2+} can affect the outcome of the fold topology (threaded or unthreaded).

We have revised the manuscript (Discussion section, p. 13) to clarify the physiological relevance of these measurements in response to this comment and others discussed below.

The threaded form has been seen before by cryoEM (R.Das lab, bioRxiv). The functional significance for these alternative structures are entirely based on speculations.

With respect, the reviewer’s assertion that the conformers differing from the threaded structure are not functionally relevant is not well-founded. First, we note that although the preprint from the Das lab (Ref. 10 in our manuscript) reported a structure for a threaded conformer, this structure was based on analysis of only a small subset of the images (~10% of all images). The survey of images of individual molecules presented in the preprint showed evidence of a large range of conformers, with many that looked significantly different from the “bent” shape of the threaded conformer and more similar to the linear shape predicted for the unthreaded conformer in Ref. 18. These other sub-populations did not have their structures analyzed in the preprint, but they are nevertheless present, indicating that this RNA forms structures other than the threaded pseudoknot conformer at significant levels, at least under the conditions used for the cryo-EM imaging.

Furthermore, with regards to the functional significance of multiple conformers in frameshift-stimulatory structures, there is substantial evidence that the conformational heterogeneity of the RNA is linked to the efficiency with which it is able to stimulate frameshifting. This evidence comes not just from studies of a wide range of stimulatory pseudoknots, but also from studies of stimulatory hairpins, as well as the effects of a small-molecule inhibitor (see, for example, Refs. 22–27 and references therein). Indeed, in the case of a virus like West Nile virus with extremely high levels of -1 PRF, there are enough different conformers of the stimulatory structure that no single one of them can be considered an “active” structure that alone stimulates -1 PRF: none of

the structures is occupied to a high enough level that it matches the level of -1 PRF. Hence static structures are insufficient on their own to account for frameshift stimulation, and the dynamic properties must be involved (see Ref. 25 and Dinman (2019) *PNAS* 116:19225–19227). It is thus clearly important and functionally relevant to understand the different structures that can be formed by the frameshift signal. The reviewer’s claim that the functional significance of alternative structures is purely speculative is thus contradicted by the literature.

We have emphasized the functional significance of the conformational heterogeneity in the revised manuscript (Discussion, p. 13).

The biggest concern is the physiological relevance -1. the FSE is taken out of its native context 2. DNA handles are placed and 3. Unfolding and refolding by optical tweezers doesn’t recapitulate unfolding by ribosome – the ribosome has a helicase and goes in sequentially along the RNA.

The reviewer’s objection here is that measurements of the pseudoknot on its own (as a fragment of the viral RNA “out of context”), in a configuration that is not identical to what happens during -1 PRF in the cell, lacks physiological relevance and hence does not provide any physiological insight. We respectfully contend that the reviewer’s conception of the type of experimental designs that allow for physiological insight and relevance is overly restrictive—by this reasoning, most of the literature on which the field bases its understanding of structure-function relationships in RNA would be judged inappropriate for providing physiological insight! In fact, studies of structure and dynamics invariably examine short fragments of RNA that define the structural feature of interest, most commonly under conditions that do not recapitulate the cellular context. Two obvious examples widely used in the literature include measurements of structures derived from x-ray crystallography (crystalline RNA does not occur physiologically), and measurements of structural dynamics using non-physiological chemical denaturants to induce conformational changes. Such studies are nevertheless routinely accepted by the field as providing insight into how RNAs function in the cell, and many of them have been published in journals like *Nature Communications* (some recent examples are listed below on p. 9).

With regards to the 3 specific sub-points mentioned by the reviewer, we address each one separately below:

1. As noted already, most studies of structure and dynamics in RNA—widely used in the field to obtain insight into physiological function—are based on short fragments of RNA taken out of the context of the larger RNA molecule in which they are found. This general rule typically applies regardless of the experimental method used, and our work here is no exception. The reason that studies focus on the relevant short fragment is that it is experimentally impractical to work with the complete RNA molecule. Looking specifically at viral RNAs, all the high-resolution structures of which we are aware in the structural databases were done using short RNA fragments containing the specific RNA of interest, “taken out of the native context” in the words of the reviewer. The same is true of all studies of structural dynamics (folding) of which we are aware, whether using ensemble or single-molecule approaches. Our work lies squarely within the widely accepted approach of studying short functional elements (in this case the frameshift-stimulatory pseudoknot), and it is unreasonable to insist that this approach makes the work physiologically irrelevant.

2. Regarding the notion that the presence of the DNA handles prevents our results from having any physiological relevance, we note that every *in vitro* experiment of structural dynamics

involves some form of modification and/or perturbation to the molecules being studied in order to monitor their behavior as they undergo the processes being studied. In some measurements, these modifications may involve attaching external dye labels to the RNA or replacing certain bases with fluorescent base analogs, in others they may involve non-physiological changes to the environment (*e.g.* chemical denaturation, large changes in temperature or ion concentrations). The question in every case should be not whether the molecule has been modified or perturbed, but rather if the modifications/perturbations change the behavior of the RNA in some significant way.

In our work, the relevant modification is the annealing of single-stranded DNA handles to the RNA to create a duplex. This duplex is required for the measurement because: (i) the pseudoknot needs to be distanced from the beads in the optical traps to prevent potential interactions with the bead, leading to the need for ~kb-long handles of some kind; (ii) if we just used ssRNA as handles, without making the handles into a duplex, the handles would form a plethora of secondary structures of varying stability, which would make it impossible to determine which unfolding/refolding events came from the pseudoknot and which came from the handle; and (iii) the handles need to be sufficiently stiff that we can detect unfolding/refolding events at relatively low force (~5 pN), which is best done with a duplex handle. The construct we used as the “baseline” for normal folding of the pseudoknot has the duplex located 6 nt away from the pseudoknot, which should be sufficiently far away that the duplex affects the folding minimally.

In order to verify that the duplex handle has little to no effect on the folding when located 6 nt away from the pseudoknot, we made a new construct in which the handle duplex was moved even further away to create a 12-nt single-stranded spacer between the handle and the pseudoknot. As described above in response to point 2 of reviewer 1, we found that the full-length unfolding curves were divided ~90%/10% into threaded/unthreaded conformers, the same (within the error of ~2–3%) as for the 6-nt spacer. This result shows that once the duplex is moved at least 6 nt away from the pseudoknot, the distance of the duplex ceases to change the pseudoknot folding, even though the duplex *does* affect the folding when it is only 1 nt away (as seen from comparing Figs. 2 and 3). Hence the handle has minimal to no effect on the folding in our baseline measurements of the folding, and it is unreasonable to assert that the use of duplex handles renders the results meaningless.

3. Turning to the objection that unfolding by optical tweezers does not precisely recapitulate unfolding by the ribosome, we agree that the correspondence between our measurements and unfolding by the ribosome is not exact. However, we respectfully dispute the reviewer’s contention that an exact correspondence, in all details, is required in order to draw relevant conclusions. From a big-picture perspective, the tweezers assay captures several physiologically important features of unfolding by the ribosome: (i) the RNA structure is actively destabilized by mechanical tension (Ref. 28), (ii) the tension ramps up/down as the ribosome tries to unfold the RNA (Ref. 30), and (iii) the RNA undergoes multiple unfolding/refolding cycles as multiple ribosomes move through the frameshift signal (Ref. 49). Some of the details are definitely different, however: as the reviewer points out, the ribosome applies force in a different geometry, pulling on the 5’ end only (instead of the 3’ end, too, as in the tweezers). The force profile over time is also more complex than the constant ramp used in our experiments, as the ribosome can apply tension for sustained periods while it is paused at the pseudoknot. These differences should of course be kept in mind, as they may change some of the details of the results, such as the values of the unfolding forces (which depend on both the pulling geometry and the profile of the

applied force over time). Nevertheless, the use of tweezers to unfold/refold the RNA is not expected to affect key results such as:

- The presence of multiple fold topologies (their existence depends on the folding mechanism).
- The pseudoknot folding mechanism (the order we found, stem 1 first/stem 2 last, is precisely the same in which the stems emerge from the ribosomal exit tunnel during translocation).
- The different mechanical rigidity of the different conformers (these are characteristics of the fold topologies).
- The role of Mg^{2+} in enhancing threading and pseudoknot rigidity (which does not depend on the ribosome).

Hence the differences between how the tweezers unfold the RNA and how the ribosome does so do not preclude us from drawing conclusions that are meaningful for how the frameshift signal behaves in the cell.

We have revised the manuscript to discuss the physiological relevance of the assay more fully (Discussion p. 10 and p. 13).

The authors clearly show that the DNA handle (which is not physiological) affects the structures.

With respect, this statement misinterprets the import of our measurements using different handle lengths. The fact that the handles on some of the constructs interfered with the 5' end was a **deliberate** part of the experimental design—the purpose was to test our hypothesis about the nature of the states N and N' by showing that we could alter their proportion in a controlled way. Through these measurements, we aimed to demonstrate explicitly that these states differed in their dependence on the behavior of the 5' end of the pseudoknot, namely whether it was threaded or unthreaded. The DNA handle only affected the folding when it was placed very close to the pseudoknot, *i.e.*, in the constructs where the handle was located 1 or –2 nt from the pseudoknot.

For the construct with the DNA handle separated from the pseudoknot by a 6-nt spacer (which has a contour length of 3.5 nm), we would expect that the handle is far enough away that it does not influence the folding and can effectively be ignored. To demonstrate more clearly that the DNA handle has little influence when far enough away from the pseudoknot, we re-measured the folding using a construct containing a 12-nt spacer between the DNA handle and the pseudoknot, as discussed above on p. 2 in response to point 2 of reviewer 1. We found that the results were identical (within error) as when the handle was 6 nt from the pseudoknot—in other words, the results don't change once the duplex is 6 nt or more away, implying that the presence of the handle can effectively be ignored when they are at that distance or greater.

We have revised the manuscript to present these new data and discuss their significance (Results p. 6, Fig. S3, and Discussion p. 13).

Overall, this is an interesting and well performed biophysical study of a short RNA piece from SARS-Cov2 but because the physiological relevance of the findings is entirely unclear I would not recommend for publication in Nature Communications.

For the reasons outlined above, we disagree that a “biophysical study of short RNA pieces” cannot provide physiological insight into RNA function, and we contend that there is clear

physiological relevance of our findings that the reviewer has overlooked. We note that many such studies have been published in *Nature Communications*, indicating that the reviewer's view of such studies is out of step with the literature. Just a few examples from the last two years include:

- Hua *et al.* (2020) *Nat Commun* 11:4531, studying the folding of a folate stress-sensing ZTP riboswitch with single-molecule (sm) FRET
- Steffen *et al.* (2020) *Nat Commun* 11:104, studying a RNA hairpin/oligo system that mimics the tertiary interactions crucial to self-splicing of the group II intron, using smFRET
- Niu *et al.* (2020) *Nat Commun* 11:5496, studying the translocation of exoribonuclease-resistant RNAs through nanopores as a proxy for their resistance to unfolding by RNases
- Holmstrom *et al.* (2019) *Nat Commun* 10:2453, studying how nucleic acid hairpin folding is assisted by disordered RNA chaperones with smFRET
- Suddala *et al.* (2019) *Nat Commun* 10:4304, studying the folding of a Mn^{2+} -sensing riboswitch with smFRET
- Mitra *et al.* (2019) *Nat Commun* 10:4318, studying the folding of fluorogen-binding RNA aptamers with SMFS

This list is not comprehensive (many more such studies have been published in *Nature* journals and other similar general-interest venues), but it shows that the field accepts the physiological relevance of the results of such studies.

Reviewer #3:

In this paper, Woodside and collaborators investigate the RNA pseudoknot of Sars-Cov-2 virus using force spectroscopy finding two distinct folded structures (N and N') with different kinetic stabilities. N and N' cooperatively unfold in a single rip where the approximately 70 nucleotides of the RNA release an extension of 35nm. The two structures can be discriminated from the bimodal rupture force distributions that show that N has larger kinetic stability and brittleness as compared to N'. The authors claim that N,N' correspond to the threaded and unthreaded conformations of the RNA pseudoknot depending on which side the 5' end of the RNA passes through the stem1/stem3 junction. To support this hypothesis additional experiments are carried out by extending the hybrid RNA/DNA 5' handle to occlude the junction, using antisense oligos to inhibit stems 1 and 3, and test the results without magnesium. In my view the conclusions are well supported by the findings, the results are well described and the text and figures clear. Overall this is a beautiful and elegant paper that I am happy to recommend for publication after the authors have considered the following (general) comments.

We thank the reviewer for these positive comments.

1. In the intro, page 3, four lines before the Results section. "Stem 1 usually folds first, followed by stem 3 and lastly stem 2" I think it should be the other way around, " ...Is there any energetic or kinetic explanation for this preferred S1-S2-S3 folding pathway?

We are unsure why the reviewer is under the impression that stem 2 folds before stem 3, as the refolding curves (as in Fig. 2c) clearly show intermediates that have lengths corresponding to the folding of stem 1 (state I_{11}) first, and then stem 1 + stem 3 (state I_{12}), before the full pseudoknot

refolds, indicating that stem 2 folds last. These results were also verified by the measurements using anti-sense oligos (as in Fig. 4), leading to the picture of the folding mechanism that is summarized in Fig. 5.

Examining the thermodynamic stability of the stems using mfold, we find that stem 1 is predicted to be the most stable, followed by stem 3, whereas stem 2 is predicted to be the least stable. The ordering of the thermodynamic stabilities therefore matches precisely with the order in which the stems form. Intriguingly, H-type pseudoknots with 2 stems are generally expected to form their stems in the order of their energetic stability (Cho *et al PNAS* (2009) 106:17349, Roca *et al PNAS* (2018) 115:E7313); our results suggest that this picture applies to this 3-stem pseudoknot, too.

We have revised the manuscript (Discussion section, p. 9–10) to discuss this issue.

2. The focuses on structures N and N', but what can be said from the other 20% second class of alternative structures? Are there any predictions for these? Does the 20% of alternative structures change in the absence of magnesium? Are these misfolded structures. In such a case I would expect a reduction of this 20% fraction in the absence of magnesium (as magnesium specific binding stabilizes native by rescuing misfolded structures).

The unfolding length and force for the conformer we denote as “Alt” tell us that (1) this structure involves 46 ± 2 nts (from ΔL_c), and (2) it contains only secondary structure (from fitting of the unfolding force distribution, which yields $\Delta x^\ddagger = 4.5 \pm 0.8$ nm, more characteristic of secondary structures like hairpins and inconsistent with the presence of tertiary contacts). Furthermore, from the fact that Alt is very slow to reconfigure into the pseudoknot (implying that it must unfold substantially or completely before it can do so), we deduce that it must involve stems that differ substantially from stems 1–3 of the pseudoknot. These considerations led us to the proposal for the structure of Alt shown in Supplementary Fig. 4c of the revised manuscript. Intriguingly, we found a very rare example of Alt converting into the pseudoknot at low force during an unfolding curve (meaning that the RNA molecule refolded first into Alt, remained in Alt during the waiting time between pulls, but changed into the pseudoknot part-way up the subsequent unfolding curve). We have added this curve to the revised manuscript as Supplementary Fig. 5.

With regards to the effect of Mg^{2+} ions on Alt, we found that the incidence of Alt was greatly reduced in the absence of Mg^{2+} , down to only 2% of curves. Because Alt is less stable energetically than N/N', this result suggests that Mg^{2+} ions kinetically trap the RNA in Alt. Looking at our model of Alt, we note its stem contains several bulges near the loop, which should destabilize it considerably under tension. As a result, we might normally expect it to reconfigure into intermediate I_1 with the native stem 1 during the refolding, leading to low occupancy of Alt (as seen in the absence of Mg^{2+}). Presumably the Mg^{2+} ions act to stabilize the stem of Alt, increasing its abundance in the conformational ensemble.

We have revised the manuscript to discuss the effects of Mg^{2+} on Alt and how they make sense in view of our model of the structure of Alt (Discussion, p. 12).

3. In the schematics of the RNA structure in Figure 2B it is not clear if this is the threaded or unthreaded structure. I presume it can be both but this should be said in the caption.

The cartoons in the insets of the figures were intended mainly to show the length of the DNA

handle, not whether the pseudoknot was threaded or unthreaded. We have revised the cartoons to emphasize the DNA handle more clearly, and we have revised the figure legends to indicate more clearly what the inset cartoon represents.

Then in the black pulling curves in the leftmost set, there are also curves showing the intermediate II. Is this the same selected set of magenta curves that are shown in the middle set of FEC?

We agree with the reviewer that this figure is somewhat confusing—the magenta curves were indeed duplicated. We have revised the figure and its legend to clarify the confusion, by showing only one set of curves with full-length unfolding and coloring the FECs in this group that have intermediates in magenta. We have also revised the figure legend to explain what the red and blue curves represent in Fig 2c, which we neglected to do previously.

4. The part on experiments without magnesium is interesting because it shows that, although magnesium further stabilizes N with respect N', it is not essential for the existence of N. Could the authors say something about the specific magnesium contacts in the stem1-stem 3 junction? Has the structure of the RNA pseudoknot been resolved (e.g. with cryo-X ray diffraction or NMR) and the magnesium binding sites been determined? What are the conformational changes induced by magnesium specific binding? Do these conformational changes occur close to the stem1-stem 3 junction? In the conclusions the authors discuss differences with the Zika virus xrRNA where threading is abolished in the absence of magnesium. How the conformational changes induced by magnesium in the RNA pseudoknot of the Zika virus differ from Sars-Cov-2?

We agree with the reviewer that these questions are interesting, and we wish we could answer them. Unfortunately, there is no structural information about the coordination of Mg^{2+} in this pseudoknot (nor indeed in any coronavirus pseudoknot), because no Mg^{2+} ions have been resolved in either of the cryo-EM studies and no other high-resolution structural methods such X-ray crystallography or NMR have been successfully applied to determine the structure of any coronavirus pseudoknot. Furthermore, no cryo-EM structures are available in the absence of Mg^{2+} , to indicate how the structure changes with Mg^{2+} binding. Hence it is difficult to address these questions.

We had assumed that, as for the Zika virus xrRNA, Mg^{2+} would be absolutely required to ensure threading, by helping to hold the 5' end in place in the stem 1/stem 3 junction while waiting for S2 to fold up. Our results show that Mg^{2+} is *not* actually essential, although it does clearly enhance the threading, given that the N population is much reduced in the absence of Mg^{2+} . Presumably this assistance with coordinating the threading of the 5' end involves Mg^{2+} ions located near the stem 1/stem 3 junction, such that they interact with both the 5' end and stem 1 and/or stem 3, but we don't know for sure if these interactions involve specific nucleotides (as opposed to non-specific interactions) nor, if so, which ones. Computational simulations suggest that Mg^{2+} ions mediate interactions between the 5' end and stem 1 that make the 3-helix junction more compact (Ref. 18), but the results from these simulations have not been confirmed experimentally.

In addition to altering the N/N' population ratio, Mg^{2+} ions make the pseudoknot structures more rigid, as we discussed in the original manuscript. Because of the fact that rigidification is observed in both the threaded and unthreaded conformers, we had hypothesized that it is especially important in the stem 2-loop 1 region, which both modeling and experiments suggest

have a dense network of tertiary contacts that might be stabilized by Mg^{2+} . Given the discussion above about the effects of Mg^{2+} on the 3-helix junction seen in simulations, and the fact that the rigidification is more extreme for the threaded conformer (4-fold decrease in Δx^\ddagger instead of 2-fold), the compaction of the junction by Mg^{2+} presumably also contributes to the rigidification of the latter.

With regards to the comparison to the Zika virus xrRNA, we note that no high-resolution structure has been reported in the absence of Mg^{2+} for this xrRNA. However, optical tweezers experiments (Ref. 21) show that in the absence of Mg^{2+} it forms secondary structure only or else a non-native unthreaded pseudoknot. Mg^{2+} ions thus have a much more pronounced effect on the Zika virus xrRNA than on the SARS-CoV-2 frameshifting pseudoknot, changing the nature of the fold instead of mainly rebalancing the folding between the two pseudoknot conformers present in the absence of Mg^{2+} .

We have revised the Discussion section to explain the effects of Mg^{2+} more clearly (p.11) and to expand on the comparison to the Zika virus xrRNA (p. 11–12).

5. The authors analyze the different kinetic stabilities of N and N' by calculating the distance to the transition state x_{++} and the kinetic barrier DG_{++} . However, it would be interesting to say something about the thermodynamic stabilities of the two states. Although N is kinetically more stable than N', it is reasonable to expect similar thermodynamic stabilities for the threaded and unthreaded conformations. In my view, the different topologies of the threaded and unthreaded conformations would mostly change the conformational entropy of the 5' end and this difference should not amount much to the free energy. Moreover a similar thermodynamic stability N vs N' would be also consistent with a regulatory feature of the PRF of the RNA pseudoknot. Could the authors derive equilibrium free energies from their data? I am aware that the FEC are irreversible but there are powerful methods to extract the free energies in pulling experiments based on classifying (from the rupture force histograms, Fig. 2D) the unfolding trajectories into N and N' states and using a fluctuation theorem for kinetic states (e.g. Alemany et al., Nat. Phys. 8 (2012) 688-694).

We agree with the reviewer that this analysis would add some important insight into the differences between the conformers. However, it is not so straightforward as proposed, because of the fact that we cannot unambiguously assign any given pulling curve to N or N': as mentioned above in response to reviewer 1, there is no significant difference between ΔL_c values for N and N', and in every measurement condition there is a significant overlap in the force distributions for the two populations. Because calculations of free energies from non-equilibrium work distributions depend very heavily on the low-energy tail of the distribution, we need to be able to define the work distributions for the two populations quite reliably over the whole range of work done (and hence unfolding forces seen).

That said, we can make an estimate of the most likely free energies for N and N' using a probabilistic approach, since we know the probability for a curve with a given unfolding force to be in N vs N', based on the fits for the unfolding force distributions. We took a bootstrap-like approach, assigning each curve to N or N' randomly, weighted by the probabilities derived from the unfolding force distributions, and then calculating the free energy for complete unfolding of N and N' for this sampling of the curves using the Jarzynski equality. We then resampled the curves into N/N' sub-populations and recalculated ΔG , repeating 5,000 times to characterize the range of values returned by the sampling procedure. We then took the average of the results as

our best estimate of ΔG . We did this separately for the curves measured with the handles located 1 or 6 nt from the pseudoknot, because for these curves the handles should not affect the pseudoknot stability, even if they change the relative proportions of N and N'. We found that the results were the same (within error) for the two different handle lengths, as expected, and the energies of the two states were very similar: the stability of N was $61 \pm 7 k_B T$ whereas that for N' was $53 \pm 6 k_B T$. The uncertainty from the Jarzynski estimates was relatively high, because there was quite a bit of dissipated work (the unfolding was not close to equilibrium). As a result, even though nominally N is suggested to be a bit more stable, this difference is not statistically significant, and $\Delta\Delta G$ is consistent with 0. Repeating the same analysis for the data without Mg^{2+} , where the unfolding is much closer to equilibrium, we found very similar energies for N and N': $54 \pm 2 k_B T$ and $52 \pm 2 k_B T$, respectively, suggesting that the two conformers do indeed have very similar thermodynamic stabilities as suggested by the reviewer.

We have revised the manuscript to include this analysis of the relative stability of the two conformers at the end of the Results section (p. 8–9).

6. I recommend calling figure 5 while showing and discussing figure 2 to better understand the experimental results of Figure 2 and better understand the threading vs non-threading differences.

Because Fig. 5 incorporates the results not just from the measurements shown in Fig. 2 but also those in Figs. 3 and 4, we believe that it would be inappropriate to present it near the beginning of the paper. We feel that the proposed approach would bias the presentation of the data, imposing our interpretation before all of the data that supported it had been presented. However, we agree with the reviewer that it would be useful to make the interpretation of Fig. 2 clearer. We have therefore revised Fig. 2 to include cartoons of the threaded and unthreaded folds that are proposed to correspond to the two peaks in the unfolding force distribution.

Most significant changes to manuscript:

- Abstract: Added statement about role of Mg^{2+} in folding, response to reviewer 3.
- Results p. 4–7, Figs. 2 and 3: Swapped the order in which the results from the constructs with different handle lengths were presented and displayed, in response to reviewers 1 and 2.
- Results p. 4, Fig. S1: Added figure showing unfolding force distribution for Alt, in response to reviewer 3.
- Fig. 2: Modified the cartoons of the pseudoknot to illustrate the threaded vs unthreaded conformers, in response to reviewer 3.
- Results p. 6: Expanded the explanation for changing the handle location, in response to reviewers 1 and 2.
- Results p. 6 and Fig. S3: Added data showing that spacers 6-nt and longer don't change the folding, in response to reviewers 1 and 2.
- Results p.8: Mentioned that the absence of Mg^{2+} disfavored Alt, in response to reviewer 3.
- Results p. 8–9: Added analysis of thermodynamic stability of pseudoknot conformers, in response to reviewer 3.
- Discussion p. 9: Mentioned that cryo-EM images showed evidence of conformers that could be unthreaded, in response to reviewer 2.
- Discussion p. 10: Explained that the order of folding matched the thermodynamic stability of the stems, in response to reviewer 3, and that it matched the order expected physiologically after ribosome passage, in response to reviewer 2.
- Fig. 5: Clarified which stem folds/unfolds at which step in the pathway, in response to reviewer 3.
- Discussion p. 11: Expanded on the effects of Mg^{2+} on the folding and structure, in response to reviewer 3.
- Discussion p. 12: Clarified the comparison to the Zika virus xrRNA, in response to reviewer 3.
- Discussion p. 12, Fig. S5: Added discussion of the properties of Alt, in response to reviewer 3.
- Discussion p. 13: Added discussion of what aspects of SMFS assay are similar to the physiological situation, what aspects differ, and how these differences might affect the results, in response to reviewers 2 and 3.
- Methods p. 17: Added details of statistical analysis of 2-population fits to force distributions, which had inadvertently been omitted.
- Methods p. 17–18: Added description of stability calculations for pseudoknot conformers, in response to reviewer 3.
- Supplementary information: Added table of pseudoknot construct sequences.
- Figs. 2–4: statistical details added to figure legends, to conform with editorial guidelines.

REVIEWERS' COMMENTS

Reviewer #1 (Remarks to the Author):

The revision looks fine to me although I still have a comment for the authors:

I do not agree with the justification provided by the authors that studying one molecule unfolding multiple times is more relevant than studying many "first pulls", from a biological point of view, given that one pseudoknot will be unfolded multiple times in the ribosome. This is certainly true, as it is that many pseudoknots will be unfolded for the first time in the same ribosome. I understand that there is a technical challenge, and that grabbing one molecule and stretching it many times is far easier than accumulating many trajectories from different molecules. But this second scenario is also relevant and authors should have seriously considered it, otherwise they only see one side of the reality.

Reviewer #2 (Remarks to the Author):

The authors did not address my criticism to show further evidence for biological relevance. Instead they provided a list of previous work that has been done in the field and suffers from the same flaws. I disagree with this type of logic- just because something has been published previously, does not mean it was correct or of broad interest to the field, or physiologically relevant. In the absence of additional experiments, I do not recommend this manuscript for publication in Nature Communications.

Reviewer #3 (Remarks to the Author):

I am satisfied with the clarifications made by the authors. My only concern is the new paragraph added (just before the discussion) on the measurement of the free energies of states N and N' for which simple Jarzynski cannot be used (because the system does not populate N and N' in equilibrium) and for which a correction term (proportional to $kT \log(\phi)$ with ϕ the fraction of misfolding) must be added (see the reference I gave in my report).

Response to reviewer comments on revised manuscript

Reviewer 1:

The revision looks fine to me although I still have a comment for the authors:

I do not agree with the justification provided by the authors that studying one molecule unfolding multiple times is more relevant than studying many "first pulls", from a biological point of view, given that one pseudoknot will be unfolded multiple times in the ribosome. This is certainly true, as it is that many pseudoknots will be unfolded for the first time in the same ribosome. I understand that there is a technical challenge, and that grabbing one molecule and stretching it many times is far easier than accumulating many trajectories from different molecules. But this second scenario is also relevant and authors should have seriously considered it, otherwise they only see one side of the reality.

We regret if we gave the impression in our last response that we did not take seriously the reviewer's suggestion to consider how the first unfolding events measured on each molecule differ from those that occur later in the series of pulling measurements—that was certainly not our intention. As we described in the previous response, we did examine the first pulls for each molecule, but the amount of information that we could obtain from the analysis was limited owing to the limited statistics: we were able to discern a significant reduction in the number of events with alternate conformers, but it would require 20–50 times as many molecules (yielding 500–1000 'first pulls') to define the unfolding force distribution sufficiently well for it to be possible to determine with any confidence if the proportions of N and N' were different. Obtaining such a large number of molecules (an order of magnitude larger than most published studies of folding using optical tweezers) would require years of additional measurement, and is therefore impractical.

With regards to the question of the biological relevance of the first time the pseudoknot is unfolded by a ribosome as opposed to subsequent ribosome-induced unfolding after the pseudoknot has refolded (which map onto first pulls and subsequent pulls in the SMFS measurements), we agree with the reviewer that both are relevant, given that every mRNA that is translated must undergo a first unfolding. However, the relative contributions of such 'first unfolding' versus 'unfolding after refolding' to the statistics of the overall behavior of the pseudoknot should presumably depend on the number of times a given mRNA molecule is translated before it is degraded. Estimates for eukaryotic cells suggest that eukaryotic mRNAs are translated on the order of 100 times or more before they degrade (Milo & Phillips, *Cell Biology by the Numbers*, Garland Science, 2015); the results might differ somewhat in the case of infection by a virus, but the first-time unfolding of a given pseudoknot by a ribosome is still almost certainly a small percentage of the total number of times that same pseudoknot is unfolded by the ribosome during translation, and thus we suspect that 'unfolding after refolding' behavior will likely dominate statistically.

Please note that we did not make any claims in the manuscript that 'unfolding after refolding' is more biologically relevant than 'first unfolding', we just noted that the former is indeed relevant physiologically. We therefore did not make any revisions to the manuscript in response to this comment.

Reviewer 2:

The authors did not address my criticism to show further evidence for biological relevance. Instead they provided a list of previous work that has been done in the field and suffers from the same flaws. I disagree with this type of logic- just because something has been published previously, does not mean it was correct or of broad interest to the field, or physiologically relevant. In the absence of additional experiments, I do not recommend this manuscript for publication in Nature Communications.

We respectfully note that—contrary to the reviewer’s claim—we did indeed highlight a number of ways in which the experimental approach was relevant biologically. We are disappointed that the reviewer did not find our work convincing, but given that the reviewer rejects the relevance of a large body of important work in the field for similar reasons, there is not much that can be done in response.

Reviewer 3:

I am satisfied with the clarifications made by the authors. My only concern is the new paragraph added (just before the discussion) on the measurement of the free energies of states N and N' for which simple Jarzynski cannot be used (because the system does not populate N and N' in equilibrium) and for which a correction term (proportional to $kT \log(\phi)$ with ϕ the fraction of misfolding) must be added (see the reference I gave in my report).

We thank the reviewer for pointing out this oversight in our calculation. We have added in the correction term, it resulted in a small change of some of the free energies. The manuscript has been revised in the Results section (p. 8) to include the corrected free-energy results and in the Methods section (p. 18) to explain how the calculation was done.